# Elevated Expression of CCN3 in Articular Cartilage Induces Osteoarthritis in Hip Joints Irrespective of Age and Weight Bearing

**DOI:** 10.3390/ijms232315311

**Published:** 2022-12-04

**Authors:** Kazuki Hirose, Miho Kuwahara, Eiji Nakata, Tomonori Tetsunaga, Kazuki Yamada, Kenta Saiga, Masaharu Takigawa, Toshifumi Ozaki, Satoshi Kubota, Takako Hattori

**Affiliations:** 1Department of Biochemistry and Molecular Dentistry, Okayama University Graduate School of Medicine, Dentistry and Pharmaceutical Sciences, Okayama 700-8558, Japan; 2Department of Orthopaedic Surgery, Graduate School of Medicine, Dentistry, and Pharmaceutical Sciences, Okayama University, Okayama 700-8558, Japan; 3Advanced Research Center for Oral and Craniofacial Sciences, Okayama University Dental School/Graduate School of Medicine, Dentistry and Pharmaceutical Sciences, Okayama 700-8558, Japan

**Keywords:** hip osteoarthritis, cartilage, cellular communication network factor 3 (CCN3), senescence-associated secretory phenotype (SASP), p16, ADAMTA4/5, IL-6, TNFα, aging, Mankin score, weight-bearing, non-weight-bearing

## Abstract

Osteoarthritis (OA) occurs not only in the knee but also in peripheral joints throughout the whole body. Previously, we have shown that the expression of cellular communication network factor 3 (CCN3), a matricellular protein, increases with age in knee articular cartilage, and the misexpression of CCN3 in cartilage induces senescence-associated secretory phenotype (SASP) factors, indicating that CCN3 promotes cartilage senescence. Here, we investigated the correlation between CCN3 expression and OA degenerative changes, principally in human femoral head cartilage. Human femoral heads obtained from patients who received total hip arthroplasty were categorized into OA and femoral neck fracture (normal) groups without significant age differences. Gene expression analysis of RNA obtained from femoral head cartilage revealed that *CCN3* and *MMP-13* expression in the non-weight-bearing part was significantly higher in the OA group than in the normal group, whereas the weight-bearing OA parts and normal cartilage showed no significant differences in the expression of these genes. The expression of *COL10A1*, however, was significantly higher in weight-bearing OA parts compared with normal weight-bearing parts, and was also higher in weight-bearing parts compared with non-weight-bearing parts in the OA group. In contrast, OA primary chondrocytes from weight-bearing parts showed higher expression of *CCN3*, *p16*, *ADAMTS4*, and *IL-1β* than chondrocytes from the corresponding normal group, and higher *ADAMTS4* and *IL-1β* in the non-weight-bearing part compared with the corresponding normal group. *Acan* expression was significantly lower in the non-weight-bearing group in OA primary chondrocytes than in the corresponding normal chondrocytes. The expression level of *CCN3* did not show significant differences between the weight-bearing part and non-weight-bearing part in both OA and normal primary chondrocytes. Immunohistochemical analysis showed accumulated CCN3 and aggrecan neoepitope staining in both the weight-bearing part and non-weight-bearing part in the OA group compared with the normal group. The CCN3 expression level in cartilage had a positive correlation with the Mankin score. X-ray analysis of cartilage-specific CCN3 overexpression mice (Tg) revealed deformation of the femoral and humeral head in the early stage, and immunohistochemical analysis showed accumulated aggrecan neoepitope staining as well as CCN3 staining and the roughening of the joint surface in Tg femoral and humeral heads. Primary chondrocytes from the Tg femoral head showed enhanced expression of *Ccn3*, *Adamts5*, *p16*, *Il-6*, and *Tnfα*, and decreased expression of *Col2a1* and *-an*. These findings indicate a correlation between OA degenerative changes and the expression of CCN3, irrespective of age and mechanical loading. Furthermore, the Mankin score indicates that the expression level of Ccn3 correlates with the progression of OA.

## 1. Introduction

Hip osteoarthritis (OA) causes pain due to the degeneration of articular cartilage and limits the range of motion of the joint, which significantly reduces the quality of life of patients [1,2]. Hip osteoarthritis is classified as either “primary” or “secondary”. “Primary hip OA” occurs without certain causes, but the risk factors are age, weight, and other unknown causes. In contrast, “secondary hip OA” involves a specific trigger, such as injury, obesity, inactivity, and genetic factors [3,4]. Primary hip OA is common in Europe and the United States, whereas secondary hip OA due to acetabular dysplasia accounts for approximately 80% of all hip OA in Japan [5,6]. The pathological features of secondary hip OA include the shallow coverage of the acetabulum on the femoral head, which induces overstress on the hip articular cartilage during loading and causes hip dysplasia, leading to the onset of OA [7,8]. Current treatments for hip OA include conservative approaches such as drug therapy, rehabilitation, and surgical treatment, such as total hip arthroplasty or osteotomy; a radical and minimally invasive treatment has yet to be established [9,10]. Although knowledge of the molecular mechanisms of primary and secondary OA is being obtained, these mechanisms are not yet fully elucidated. In the hip joint, we can clearly distinguish and obtain the normal and OA femoral heads from femoral neck fracture (normal) specimens and from OA hip arthroplasty specimens; the femoral heads are unevenly weight-bearing and can be used to distinguish between weight-bearing and non-weight-bearing parts. For this reason, we believe that the hip joint has advantages in analyzing the molecular mechanisms of primary and secondary hip osteoarthritis [11].

Cellular communication network factor 3 (CCN3), previously called nephroblastoma overexpressed (NOV), is a secreted multifunctional protein involved in a variety of cellular processes, and it interacts with various extracellular and transmembrane proteins [12,13,14]. Our previous studies have shown that CCN3 suppresses the proliferation and maturation of growth plate chondrocytes, regulating the expression of chondrocytic extracellular matrix (ECM) genes and Sox9 [15]. In a recent study, we reported that the expression of CCN3 increases with age in human knee articular cartilage. Similarly, transgenic mice overexpressing CCN3 in cartilage developed degenerative changes with the induction of a senescence-associated secretory phenotype (SASP) in knee joint cartilage from an early age, indicating that aging induces CCN3, which, in turn, promotes cartilage senescence markers [16]. Here, we further investigate the correlation between CCN3 and OA using human and mouse hip joint cartilage and provide evidence to suggest that the overexpression of CCN3 causes OA. The purpose of this study is to analyze whether OA degeneration in cartilage correlates with the high expression of CCN3 independently of age or weight bearing.

## 2. Results

### 2.1. Hip Femoral Articular Cartilage from Human Patients

Human femoral heads were obtained from patients undergoing total hip arthroplasty due to OA (“OA” group, *n* = 18, male 1, female 17, age 47–82; Figure 1(A-1–A-3)), based on diagnosis at the Orthopaedic Department at Okayama University, and femoral neck fractures (“normal” group with apparently normal cartilage, *n* = 11, male 4, female 7, age 60–86; Figure 1(B-1–B-3)). For the normal group, radiographic examination confirmed that there was no irregularity of the acetabulum and femoral head, and no signs of osteosclerosis or osteophyte formation. Furthermore, normal femoral heads were examined morphologically for signs of the beginning of OA after they were removed. There was no significant difference in the age distribution between the “OA” and “normal” group (Mann–Whitney U test, *p* < 0.05; see more details in the Materials and Methods; Table 1). Cases with other complications, such as rheumatoid arthritis, osteonecrosis of the femoral head, tumors, usage of anticancer agents, oral steroids, developmental dysplasia of the hip, and other autoimmune diseases, were excluded. Articular cartilage was obtained from both weight-bearing parts (above the head, indicated by the yellow arrow in Figure 1C) and non-weight-bearing parts (lower anterior to the head, blue arrow).

### 2.2. Safranin-O Staining and Mankin Score Evaluation of Degeneration Grade of Human Femoral Cartilage

Weight-bearing and non-weight-bearing parts of human femoral surface tissues, as indicated in Figure 1, were collected and stained with Safranin-O-fast green, and the degenerative changes in cartilage were evaluated and scored using the Mankin score (Figure 2A–D) [17,18]. In both the weight-bearing and non-weight-bearing group, the Mankin score in the OA group was significantly higher than in the normal group (*p* < 0.05). Comparing the weight-bearing and non-weight-bearing parts, the Mankin score of the weight-bearing part was significantly higher than in the normal group (*p* < 0.05). In the OA group, the Mankin score in the weight-bearing group was higher than in the non-weight-bearing group (*p* < 0.05) (Figure 2E). No correlation between age and Mankin score was observed, but the Mankin score was distributed into three distinct OA and normal groups (Figure 2F).

### 2.3. Elevated Gene Expression of CCN3 and OA-Related Factors in Osteoarthritic Human Hip Femoral Articular Cartilage

For the semi-quantitative analysis of the gene expression of *CCN3*, OA-induced factors, and SASP factors, total RNA was collected both from cartilage pieces and from primary cultures of chondrocytes. In the analysis of cartilage, elevated *MMP13* expression, as well as *CCN3,* was observed in the non-weight-bearing group of OA compared with the normal group (*p* < 0.05), but not between the weight-bearing groups (Figure 3A). The expression of cartilage hypertrophy marker *COL10A1* was higher in the weight-bearing OA group compared with the normal group (*p* < 0.05, Figure 3A). Comparing the weight-bearing part with the non-weight-bearing part, the expression of *COL10A1* was higher in the weight-bearing part in the OA group (*p* < 0.05, Figure 3A). In contrast, in human primary chondrocytes, significantly elevated expression of *CCN3* was observed in the weight-bearing OA group versus the normal group (*p* < 0.05, Figure 3B). However, in the non-weight-bearing group, *CCN3* expression was elevated in OA compared with the normal group, but it was not significant. Similarly, OA-related degenerative factors such as *ADAMTS4*, SASP factors such as *IL-1β*, and cell cycle arrest genes such as *p16* showed elevated levels in the OA weight-bearing group compared with the normal weight-bearing group (*p* < 0.05, Figure 3B). Additionally, we observed the significantly enhanced expression of *ADAMTS4* and *IL-1β* in the non-weight-bearing group of OA compared with the normal group (*p* < 0.05, Figure 3B). However, no significant difference was observed between the weight-bearing and non-weight-bearing groups among OA or normal groups themselves in all the genes examined, except for *p16* expression in the normal group (Figure 3B).

### 2.4. Immunohistochemical Analysis of CCN3, Aggrecan Neoepitope, MMP13, and ADAMTS4 in Human Femoral Articular Cartilage

To determine the relationship between CCN3 and cartilage degeneration at the protein level, human femoral surface tissues were immunostained with anti-CCN3 antibody. CCN3-positive signals were strongly observed in chondrocytes tangential to the transitional zone, as well as in the matrix of the roughened surface of articular cartilage in both the weight-bearing part and non-weight-bearing part of the OA group, but more CCN3-positive cells were observed in clustered chondrocytes, and a more roughened surface was shown in the weight-bearing part of the OA group. A positive but faint CCN3 signal was observed in the normal group, for both weight-bearing and non-weight-bearing parts (Figure 4A). Marker products of cartilage degeneration, aggrecan neoepitope (Figure 4B), MMP13 (Figure 4C), and ADAMTS4 (Figure 4D) were also detected in the OA group, especially in clustered chondrocytes tangential to the transitional zone of the weight-bearing part and the matrix of the roughened surface of the articular cartilage in non-weight-bearing parts. No staining was observed in the normal group (Figure 4B–D). These data indicate that the enhanced strong accumulation of CCN3, cartilage degradative enzymes, and degradative products of articular cartilage were observed in osteoarthritis patients irrespective of age and weight bearing, even though chondrocytes did not show enhanced expression at the mRNA level. Furthermore, analyzing the level of gene expression in cartilage tissue was found to be important to monitor the expression levels of genes such as *COL10A1* and *MMP13*, which could not be reproduced in cultured cells.

### 2.5. Correlation between the Expression of CCN3 and Mankin Score

The correlation between the CCN3 level analyzed from RNA directly purified from the cartilage and the Mankin score of the adjacent tissue was examined (Figure 5). A positive correlation between the CCN3 mRNA level and tissue degeneration was obtained (Spearman’s rank correlation coefficient, *p* < 0.05, Figure 5).

### 2.6. Cartilage-Specific Overexpression of Ccn3 Induced OA-like Degenerative Changes in Particular Hip Joint In Vivo

To determine the role of Ccn3 in cartilage degeneration in vivo, we generated a mouse model that overexpressed Ccn3 specifically in the cartilage with a *Col2a1* promoter–enhancer (Tg) [19]. Since weight bearing on articular joints in mice is not the same as that in humans, we analyzed femoral heads and shoulder joints as relatively less weight-bearing joints. Immunohistochemical staining by anti-Ccn3 antibody confirmed the over-accumulation of Ccn3 in 2-month-old shoulder joint articular cartilage (Figure 6A), as well as 3-month-old hip femoral articular cartilage bearing less weight (Appendix A), in not only the tangential zone but also the whole cartilage, with a roughened surface in Tg mice (Figure 6A and Appendix A). Cartilage degeneration was also monitored by aggrecan neoepitope detected by anti-aggrecan neoepitope antibody (Figure 6C and Appendix A). Positive cells were observed not only in the superficial zone but also in the radial zone, close to the tide mark in Tg (Figure 6C). X-ray images of 2-month-old shoulder (Figure 6D) and hip joints (Appendix A) showed deformity and osteoarthritis-like degenerative changes in Tg. Primary cultured chondrocytes were isolated from two-month-old femoral head cartilage from both Tg and WT mice, and their RNAs were purified after confluency. In primary *Ccn3* Tg femoral articular chondrocytes, the gene expression of *Adamts5*, *Il6*, *Tnf*α, *p16*, and *Ccn3* was enhanced, but *Acan* and *Col2a1* were significantly downregulated (*p* < 0.05, Figure 6F). These data indicate that the overexpression of Ccn3 in articular cartilage induces osteoarthritic degenerative changes.

## 3. Discussion

The proposed risk factors of OA include aging, being female, obesity, genetic factors, environmental factors, and trauma; however, the nature of the true causes leading to drastic treatment is still controversial [9,10]. Previously, we reported that the expression of CCN3 increases with age in human knee articular cartilage. Similarly, transgenic mice overexpressing CCN3 in the cartilage developed degenerative changes with the induction of a senescence-associated secretory phenotype (SASP) in knee joint cartilage from an early age, indicating that aging induces CCN3, which, in turn, promotes cartilage senescence markers [16]. Here, we hypothesize that the high CCN3 expression could be one of the causes of OA, and we investigate the correlation between CCN3 and OA using human and mouse hip joint cartilage.

In this study, the human femoral neck samples that we collected were predominantly female, consistent with the higher prevalence of OA and fractures in females [20,21]. In addition, no significant differences in the age and body mass index were observed between the OA and fracture (normal for cartilage) groups for all of the experiments in which the gene expression was analyzed, such as human primary cultured chondrocytes and cartilage. In our previous report, we showed that the CCN3 expression level was correlated with age, and that elderly people who are prone to obesity are more likely to develop osteoarthritis [16,22]. In this study, individual differences in age and physical disparity could be omitted.

In the present study, Safranin-O staining and the Mankin score of our collected femoral cartilage showed significantly higher values in the OA group than in the normal group, suggesting that a comparison between the OA and normal groups could be achieved, which was not possible for knee joints. As previously reported, the Mankin score indicates the tissue damage associated with OA [17,18]. A comparison between the weight-bearing part and non-weight-bearing part can be also achieved by using the hip femoral head. The Mankin score of the weight-bearing part of the OA group was significantly higher than that of the non-weight-bearing part; however, immunostaining of CCN3 showed a strong positive signal regardless of the Mankin score, indicating that accumulated CCN3 is observed in both the weight-bearing and non-weight-bearing parts. Based on our data, high CCN3 accumulation is strongly observed in OA regardless of the load on the cartilage.

We evaluated gene expression from primary cultured chondrocytes and cartilage tissue without culturing to establish the changes in gene expression during culturing. The gene expression of cultured chondrocytes between OA and normal tended to be similar compared with that of cartilage; however, some genes—for example, hypertrophy markers of chondrocytes, such as *MMP13* and *COL10A1*—were not expressed in cultured chondrocytes [23]. Gene expression in the weight-bearing part and non-weight-bearing part, especially in OA samples, seems to change after culturing. For example, *CCN3* expression in RNA from the weight-bearing part of OA cartilage did not show a significant enhancement compared with that of the normal group; however, after culturing, the cells from the weight-bearing part of OA could express more *CCN3* and showed a significant difference from the normal group [24,25]. In contrast, COL10A1 expression in RNA from the weight-bearing part of OA cartilage was significantly higher compared with that from the normal cartilage. Furthermore, comparing both the weight-bearing part and non-weight-bearing part from the OA cartilage, expression was significantly higher in the weight-bearing part, which was also expected [26,27]. For critical evaluation, more careful analysis seems to be required.

In human primary chondrocytes, the significantly elevated expression of not only *CCN3* but also OA-related degenerative factors such as *ADAMTS4*, SASP factors such as *IL-1β*, and cell cycle arrest genes such as *p16* was observed in the OA weight-bearing group compared with the normal weight-bearing group [28,29]. Additionally, we observed the significantly enhanced expression of *ADAMTS4* and *IL-1β* in the non-weight-bearing OA group compared with the normal group. However, no significant difference was observed between the weight-bearing and non-weight-bearing parts in either the OA or normal groups in all of the genes examined, indicating that the levels of *CCN3* and OA-related markers were significantly higher in the OA group regardless of weight bearing. Furthermore, immunohistochemistry with anti-CCN3, anti-aggrecan neoepitopes, anti-MMP13, and anti-ADAMTS4 antibodies showed enhanced staining in the OA group regardless of weight bearing. The human OA and normal samples that we collected did not show a correlation between increased expression of CCN3 and aging in both groups (Spearman’s rank correlation coefficient, *p* > 0.05). One of the causes of osteoarthritis of the hip is acetabular dysplasia, which accounts for 80% of cases in Japan, and, due to morphological abnormalities of the hip joint, excessive mechanical stress is applied to the femoral heads [5,6,7,8]. Recent studies have reported that osteoarthritis expresses a specific gene when exposed to excessive mechanical stress [7,30,31,32,33]; however, high expression of CCN3 was observed for OA irrespective of age and weight. This indicates that the elevation of CCN3 expression leads to the enhancement of OA and confirms our hypothesis that enhanced CCN3 levels in hip cartilage may be the cause. The CCN3 level analyzed in directly purified RNA correlated positively with the Mankin score of adjacent tissue, even in normal cartilage. This is also strong evidence to suggest that the elevation of CCN3 correlates with enhanced cartilage destruction.

In this report, we generated *Ccn3* transgenic mice with Ccn3 overexpression in their cartilage, and the hip joints were analyzed by immunohistochemistry. It was revealed that aggrecan neoepitope accumulated in femoral articular cartilage, and that Ccn3 accumulated in Tg femoral articular cartilage. In addition, irregularities of the acetabulum and the femoral head were confirmed. X-ray analysis showed deformity in the Tg femoral head compared with wild-type mice. Messenger RNA was collected from primary cultured chondrocytes isolated from the femoral head cartilage of *Ccn3* transgenic mice, and an increase in *Ccn3*, an increase in OA-related marker genes, and a decrease in aggrecan and type 2 collagen genes were observed. These results suggest that the overexpression of Ccn3 in hip femoral cartilage is strongly associated with OA changes in the hip. Elevated *p16*, *Tnfα*, and *Il6* in *Ccn3* transgenic mice suggests the acceleration of cartilage senescence, which is in line with our previous report [16]. Since degenerative changes in mouse shoulder joints exhibit similar phenotypes to the hip joints [34,35], we analyzed the shoulder joints in addition to the hip joints of *Ccn3* transgenic mice and WT mice. With Safranin-O-fast green staining, OA-like degenerative changes in Tg shoulder joints, such as irregular surfaces and narrowing joint spaces, were observed. Immunohistochemical analysis revealed accumulated aggrecan neoepitope in Tg shoulder joints, as well as accumulated Ccn3. X-ray and Micro-CT analysis showed deformity in Tg shoulder joints but not in WT mice. These results are consistent with the findings obtained with human subjects with or without weight bearing.

The exact molecular mechanism of OA progression by CCN3 is still unknown. In the future, it is necessary to investigate the correlation between CCN3 and OA changes in other joints, such as shoulder joints and ankle joints in human cases. In addition, cartilage needs to be analyzed by RNA sequencing analysis to identify OA-related genes and investigate how CCN3 is involved in these genes. CCN3 has been reported to act as a suppressor of proliferative activity in tumor cell lines [36,37,38]. CCN3 has also been shown to be induced in a glucose-depleted state, such as a starvation state [39,40]. From these reports, CCN3 induction may be considered a protective response to external stimuli such as aging and starvation.

A further limitation of this study is that femoral neck fractures and osteoarthritis are diseases found in the elderly, with an average age of 70 years old, so osteoporosis cannot be ruled out.

## 4. Materials and Methods

### 4.1. Population and Sample Collection

All samples used in this study were obtained from patients (*n* = 29) who underwent total hip arthroplasty due to OA (“OA” group), and femoral neck fractures (“normal” group) at Okayama University Hospital in Okayama, Japan. Informed consent was obtained prior to sample collection from all of the patients that participated in this study.

For the evaluation of cartilage RNA, cartilage samples (4 for the normal group and 12 for the OA group) were collected, and the average age at surgery for each group was 74.0 ± 6.8 years (normal) and 71.9 ± 10.0 years (OA); the gender ratio (male:female) was 1:3 (normal) and 1:11 (OA); and the BMI was 19.8 ± 4.0 kg/m^2^ (normal) and 25.8 ± 5.3 kg/m^2^ (OA). Significant differences in distribution were not observed between the two groups in any of the parameters (Mann–Whitney U test, *p* > 0.05). For the evaluation of RNA from primary chondrocytes (4 for the normal group, and 11 for the OA group), the average age at surgery in each group was 73.5 ± 6.6 years (normal) and 70.2 ± 10.1 years (OA); the gender ratio (male:female) was 1:3 (normal) and 1:10 (OA); and the BMI was 21.1 ± 1.6 kg/m^2^ (normal) and 25.7 ± 5.5 kg/m^2^ (OA). Significant differences in distribution were not observed between the two groups in any of the parameters (Mann–Whitney U test, *p* < 0.05, Table 1). As exclusion criteria, rheumatoid arthritis, osteonecrosis of the femoral head, tumors, usage of anticancer agents, oral steroids, development dysplasia of the hip, and other autoimmune diseases were considered. In addition, if OA changes, such as narrowing of the joint space in X-ray findings, were suspected in the normal group, the sample was excluded. Since we previously reported that the CCN3 level increases with aging [16], patients aged over 82 years old were excluded to omit aging effects.

### 4.2. Treatment of Femoral Heads

Within 3 h after surgery (total hip arthroplasty or bipolar hip prothesis), femoral heads were isolated from the femoral neck with a bone saw, covered with saline-moistened gauze, and brought to the laboratory under sterile conditions. In the femoral head, the weight-bearing part (the top of the head) and the non-weight-bearing part (the anterior-inferior of the head) were selected. For histological analysis, joint surface tissue including cartilage and subchondral bone was collected, and for RNA analysis, the cartilaginous part was removed with knives from each part, defined as the “weight-bearing part” and “non-weight-bearing part”. In the OA group, severe cartilage damage could be visually observed in the weight-bearing part, but cartilage damage in the non-weight-bearing part was milder than in the weight-bearing part. In the normal group, no cartilage damage was observed in either the weight-bearing part or the non-weight-bearing part.

### 4.3. Cell Culture

Human and mouse cartilage pieces of the femoral head were washed with PBS 3 times, and soft tissues remaining on the surface were digested with 0.1% trypsin for 5 min at 37 °C. Cells released from cartilage pieces were washed with Dulbecco’s modified Eagle’s medium (DMEM) containing 10% fetal bovine serum (FBS) 3 times, and the cartilage pieces were digested with 2% collagenase overnight. The released chondrocytes were cultured in Dulbecco’s modified Eagle’s medium (DMEM) containing 10% fetal bovine serum (FBS) at 37 °C with 5% CO_2_ to reach confluence.

### 4.4. Isolation of Total RNA

Total RNA was isolated with the RNeasy Mini Kit (Qiagen, Hilden, Germany) according to the manufacturer’s instructions.

For direct RNA isolation from the cartilage, cartilage pieces were minced with a homogenizer (PRO250, PRO Scientific, Oxford, Connecticut) in ISOGEN (Nippon Gene, Toyama, Japan) and further homogenized with a glass homogenizer. According to the protocol, the sample was centrifuged after the addition of chloroform, and the supernatant was purified with the RNeasy Mini Kit.

### 4.5. Real-Time Polymerase Chain Reaction

Total RNA was reverse-transcribed using a PrimeScript RT Reagent Kit (Takara Bio, Shiga, Japan). Subsequent quantitative real-time PCR was performed using SYBR^®^ Green Realtime PCR Master Mix (Toyobo Osaka, Japan) with a StepOnePlus™ (Applied Biosystems, Basel, Switzerland) Real-Time PCR System. The expression of Gapdh was used to standardize the total amount of cDNA. The primer sequences and cycle conditions were described previously [16].

### 4.6. Histology

The human and mouse tissues were fixed with 10% formaldehyde/PBS for 36 h, followed by decalcification with OSTEOSOFT (Merck, Darmstadt, Germany) until they became soft enough to be cut with knives. The tissues were dehydrated and embedded in paraffin, and 5-µm-thick sections were prepared. To evaluate the cartilage damage, Safranin-O-fast green staining was performed.

Immunohistochemistry was conducted as reported previously [16].

### 4.7. Analyzing Hip Joints of Ccn3 Transgenic Mice

Ccn3 transgenic (Tg) mice were generated as reported previously [16].

After removing the femoral head from the sacrificed mouse, all of the muscles attached to the femur and acetabulum were removed under a magnifying glass. The cartilage on the femoral head was collected and primary chondrocytes were isolated as described above.

### 4.8. Mankin Score

The grade of cartilage destruction was assessed with Safranin-O staining and Mankin scoring [17,18].

### 4.9. X-ray Analysis

Degenerative changes in the hip joint were analyzed by examining radiographs. Immediately after sacrificing the mouse, an incision was made in the hip joint immediately above the femur to avoid the outward movement of the joint. Images of the natural flexion position of the hip joint from the side were taken under constant conditions (40 kV, 5 mA for 3 s, Fujicolor, Sofron SRO-M50, Tokyo, Japan).

## 5. Conclusions

In conclusion, in this study, it was found that the high expression of CCN3 is not related to load or aging but is correlated with osteoarthritis. The high expression of CCN3 was positively correlated with the Mankin score, and it was found that the expression of CCN3 increased as the grade of OA increased. The overexpression of CCN3 in the articular cartilage of shoulder joints induced OA-like changes, as found in hip joints.

## Figures and Tables

**Figure 1 ijms-23-15311-f001:**
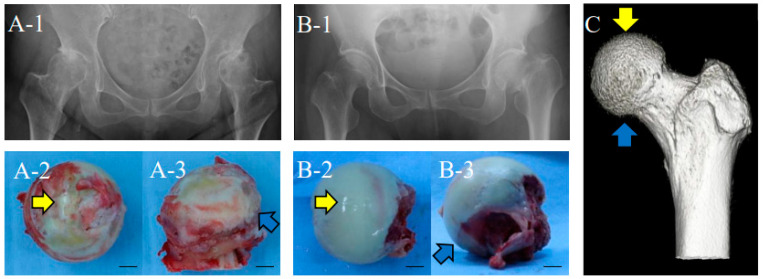
Human femoral heads obtained from patients. X-ray photographs of frontal view of hip, (**A-1**): OA; (**B-1**): normal. The left side is the affected area. Diagnosis was based on the criteria of an orthopedic surgeon at Okayama University Hospital. Samples with suspected OA findings, such as narrowing of the joint space, were excluded from the normal group composed of bone fracture patients. (**A-2**) and (**B-2**): upper view of femoral heads isolated from patients. Weight-bearing parts are indicated by yellow arrows and non-weight-bearing parts are indicated by blue arrows. (**A-3**) and (**B-3**): lateral view of the same femoral heads as (**A-2**) and (**B-2**), respectively. Non-weight-bearing parts are indicated with blue arrows. Bars: 1 cm. (**C**): 3D-constructed CT image of femoral head. The weight-bearing part is indicated with a yellow arrow, and the non-weight-bearing part is indicated by a blue arrow.

**Figure 2 ijms-23-15311-f002:**
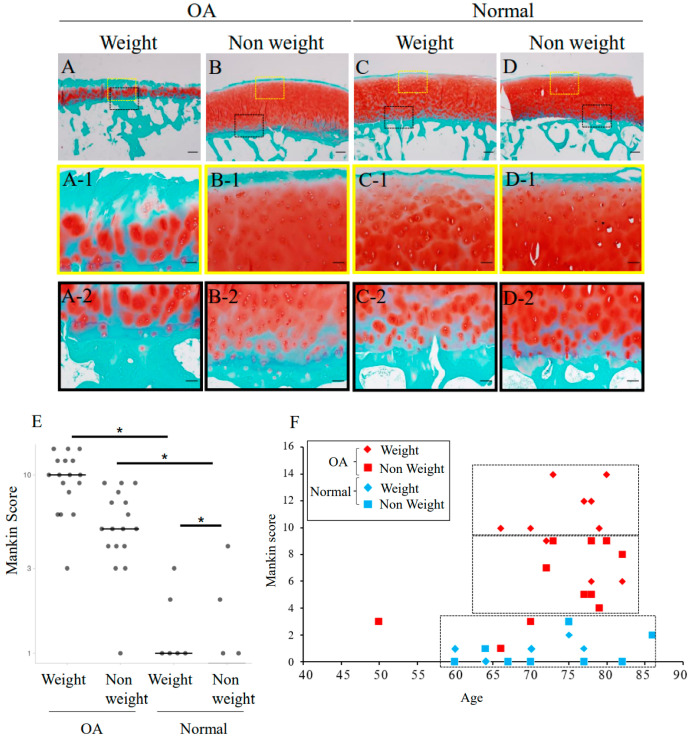
Safranin-O-fast green staining of tissues from human femoral heads. (**A**,**B**): OA; (**C**,**D**): normal. (**A**,**C**): weight-bearing parts; (**B**,**D**): non-weight-bearing parts. (**A-1**,**2**–**D-1**,**2**): magnified images indicated as yellow and black dashed boxes in (**A**–**D**). Mankin score of degeneration of cartilage was evaluated using (1) state of the joint surface, (2) morphology of chondrocytes embedded in the matrix, (3) matrix stainability by Safranin-O, and (4) visibility and continuity of the tidemark, and the total score was evaluated as the degree of cartilage degeneration. Bars: 200 µm. (**E**): Distribution of evaluated Mankin score of each sample. *: *p* < 0.05. (**F**): Relationship between Mankin score and age. No significant correlation was observed; however, the Mankin score was distributed into three distinct OA and normal groups.

**Figure 3 ijms-23-15311-f003:**
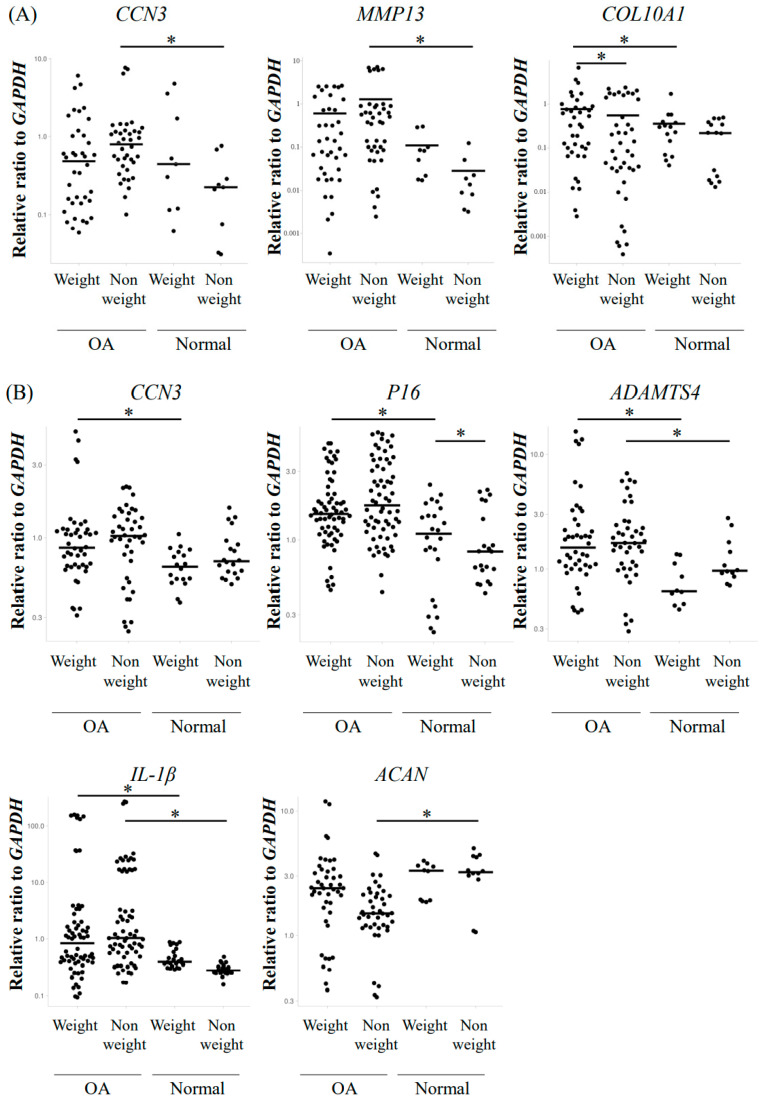
Gene expression analysis in (**A**) human hip femoral cartilage and (**B**) primary cultured chondrocytes. Both OA and normal RNA was collected from the weight-bearing part (weight) and non-weight-bearing part (non-weight) of the articular cartilage (* *p* < 0.05).

**Figure 4 ijms-23-15311-f004:**
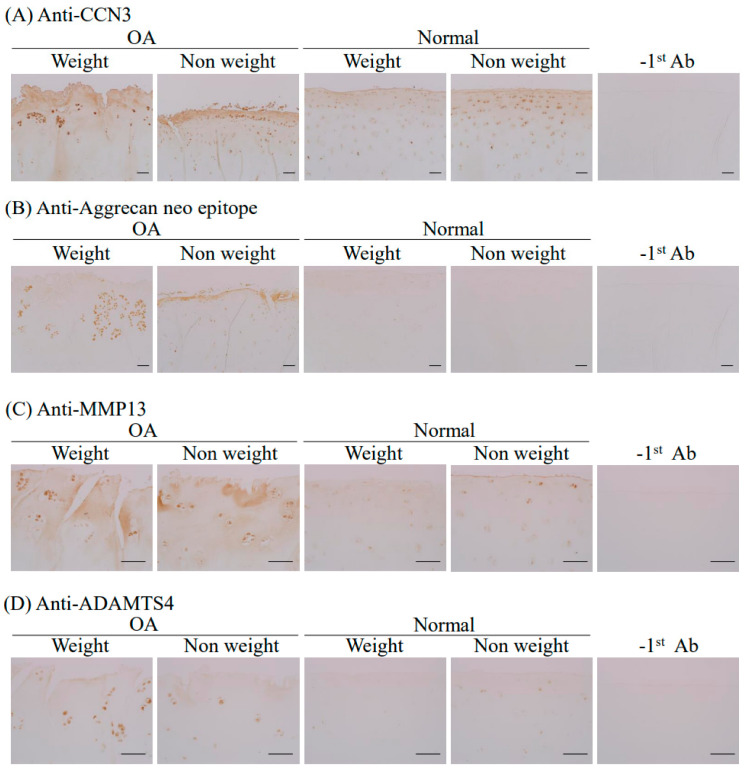
Immunohistochemical staining of (**A**) CCN3, (**B**) aggrecan neoepitope, (**C**) MMP13, and (**D**) ADAMTS4 in tissues from human femoral heads. Left 2 panels: OA group; right 2 panels: normal group. −1st Ab: negative staining control without 1st antibody. Bars in (**A**,**B**): 200 µm, and bars in (**C**,**D**): 100 µm.

**Figure 5 ijms-23-15311-f005:**
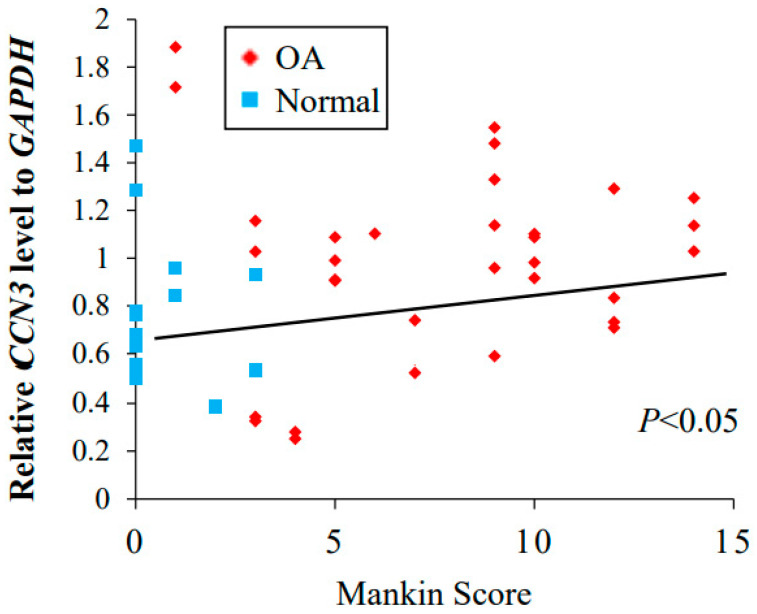
Correlation between the expression of *CCN3* and Mankin score (*p* < 0.05).

**Figure 6 ijms-23-15311-f006:**
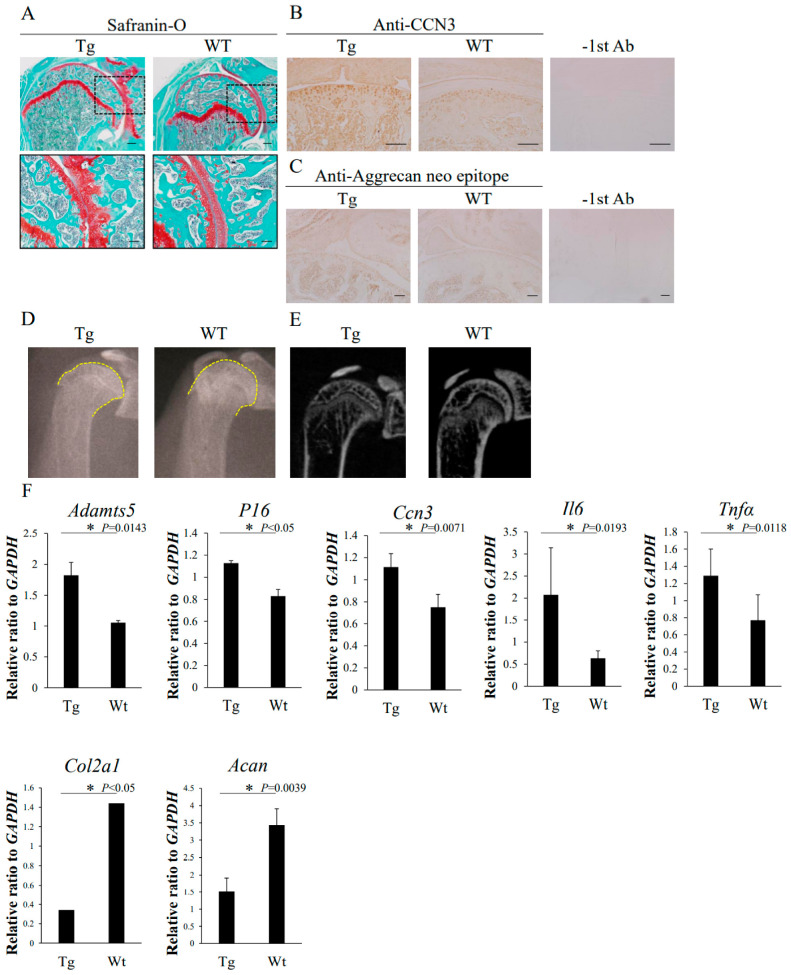
Histological analysis of shoulder joints from 2-month-old Ccn3 Tg and WT. (**A**) Safranin-O-fast green staining of Tg (left) and WT (right). Bottom photographs are magnified images of dashed boxes in upper panels. Immunohistochemical staining of (**B**) CCN3 and (**C**) aggrecan neoepitope of Tg (left) and WT (right). −1st Ab is the negative control without first antibody. (**D**) X-ray images of shoulder joints of 2-month-old Ccn3 Tg (left) and WT (right). (**E**) Micro-CT images of shoulder joints of 2-month-old CCN3 Tg (left) and WT (right). (**F**) Gene expression analysis of primary cultured femoral cartilage from 2-month-old Tg and WT. The *p*-Value is indicated for each gene, and *p* < 0.05 (*) is considered significant. Bars: 100 µm.

**Table 1 ijms-23-15311-t001:** Parameters of human hip femoral heads analyzed.

human samples for direct RNA purification from cartilage			
Variables	OA	Normal	*p* value
Males: Females	1:11	1:03	NS
Age (years)	71.9 ± 10.0	74.0 ± 6.8	NS
BMI (kg/m^2^)	25.8 ± 5.3	19.8 ± 4.0	NS
human samples for direct RNA purification from primary cultured chondrocytes			
Variables	OA	Normal	*p* value
Males: Females	1:10	1:03	NS
Age (years)	70.2 ± 10.1	73.5 ± 6.6	NS
BMI (kg/m^2^)	25.7 ± 5.5	21.1 ± 1.6	NS

## Data Availability

Not applicable.

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
