# Peer review of "Elevated Expression of CCN3 in Articular Cartilage Induces Osteoarthritis in Hip Joints Irrespective of Age and Weight Bearing"

_ijms, 2022, doi:10.3390/ijms232315311_

Round 1
Reviewer 1 Report
Dear Authors,
This is a novel and path-breaking study and I congratulate the authors for taking up this hectic task. But I am worried about a few points listed in the article. Please find my queries attached herewith.
1. Please give your reasons for the variation in cohort size, in the context wherein neck fracture surgeries are relatively common.
arthroplasty n=18/normal =11
2. Can an increase in the cohort size make the study more strong statistically? Please give your valid points in this regard.
3. A)In the group of cohorts aged over 65-70, how do the authors conclude whether the femoral head is normal or deceased?
4. A)Concerning question 3 above, whether only the appearance of the femoral head is taken into consideration in concluding?
4.B) How would the authors arrive at a decision if the appearance is at a borderline between the normal state and arthritic state?
5. Please give the rationale for including cohorts aged 80 years and above in the normal group.
6. The title of the article and the conclusions corroborate CCN3 expression to osteoarthritis, but an array of variables such as aggrecan neoepitope, MMP13, ADAMTS4,IL-1,ACAN,p16 etc IN SECTIONS 2.3,2.4 AND 2.6 ARE DESCRIBED.
This makes the paper complicated and the reader can get confounded with too many variables. Kindly edit /delete so that the article can be presented to the readers in a clear and simple way.
We look forward to hearing from you soon.
Thank you.
Author Response
Answer to the reviewers
Reviewer 1
Answer: Please see the files attached.

Reviewer 2 Report
The research fouses on highlighting eventual correlations between OA disease and the expression of CCB3 protein.
The paper is very well presented (figures are excellent), a minor comment is reported to improve the final work:
- the introduction, while briefly presenting the OA burden, lacks in detailing other studies on CCN3. Indeed, it is worth comparing in the discussion section the outcomes of the present research with other similar literature works.
Author Response
The paper is written very well, it is clear and concise. The authors explain very well in this paper about CCN3 in articular cartilage may induce osteoarthritis in hip joints irrespective of age and weight bearing.
I am recommended to publish this work without any modification.
Answer: Thank you very much for these positive comments.

Reviewer 3 Report
The paper is written very well, it is clear and concise. The authors explain very well in this paper about CCN3 in articular cartilage may induce osteoarthritis in hip joints irrespective of age and weight bearing.
I am recommended to publish this work without any modification.
Author Response
Reviewer 3
The paper is written very well, it is clear and concise. The authors explain very well in this paper about CCN3 in articular cartilage may induce osteoarthritis in hip joints irrespective of age and weight bearing.
I am recommended to publish this work without any modification.
Answer: Thank you very much for these positive comments.

Round 2
Reviewer 1 Report
Thank you for your well-explained answers.